# The Effects of Prescribed Physical and Cognitive Exercise on Life Satisfaction, Self-Efficacy and Mood States in Adults with Down Syndrome: The MinDSets Study

**DOI:** 10.3390/ijerph21050610

**Published:** 2024-05-10

**Authors:** Viviane Merzbach, Matthew Jewiss, Adrian Scruton, Dan Gordon

**Affiliations:** 1Cambridge Centre for Sport & Exercise Sciences, Anglia Ruskin University, Cambridge CB1 1PT, UK; viviane.merzbach@aru.ac.uk (V.M.); matthew.jewiss@aru.ac.uk (M.J.); adrian.scruton@aru.ac.uk (A.S.); 2Complexité, Innovation, Activités Motrices et Sportives (CIAMS), Université, Paris-Saclay, CEDEX, 91405 Orsay, France

**Keywords:** Trisomy-21, intellectual disability, physical activity, cognitive training, executive function, BrainHQ, quality of life, subjective well-being, self-worth, adaptive mood states

## Abstract

Down syndrome (DS) is characterised by a duplication of chromosome-21 and is linked to co-occurring physical and mental health conditions, including low self-efficacy and disturbed mood states. The purpose of this study was to investigate the effects of an eight-week prescribed physical and/or cognitive training intervention on measures of mood disturbance, life satisfaction and self-efficacy in a population of adults with DS. Eighty-three participants (age 27.1 ± 8.0 years) from across five continents volunteered. Participants were assigned using matched groups based upon performance in a modified six-minute walk test to either an exercise (EXE) 3 × 30 min of walking/jogging per week, cognitive training (COG) 6 × 20 min per week, a combined group (COM) or the control (CON) who did not complete any intervention. Profile of Mood States (POMS) were assessed using a five-point scale across 65 categories pre- and post-study as well as upon completion of each week of the intervention. In addition, Satisfaction with Life Scale (SWLS) and self-efficacy using the Generalised Self-Efficacy scale (GSE) were recorded before and after the intervention. GSE increased for all participants by 1.9 ± 5.2 (*p* = 0.002) from pre- to post-intervention, while POMS showed significant changes for the whole group from pre- to post-intervention for tension (*p* < 0.001), depression (*p* < 0.001) and for anger (*p* < 0.001). In addition, significant correlations were observed between SWLS and ΔTMD, Δtension, Δanger, and Δfatigue (*p* < 0.05) for EXE. Both COG and EXE provide a framework for empowering enhancements in life satisfaction, self-efficacy and mood states fostering improvements in quality of life.

## 1. Introduction

Down syndrome (DS) is a chromosomal disorder characterised by an additional duplication of either the whole or part of chromosome-21. It accounts for ~14 in 10,000 live births in the USA [1] and ~1.72 in 1000 births in the UK [2]. It is characterised by diminished cognitive and executive function [3], chronotropic incompetence [4], elevated adiposity [5], neuromuscular conditions [6], immunological suppression [7] and reduced lung function [8]. Additionally, individuals with DS express poor short-term memory [9] but show relative strength in visual–spatial awareness and sensory processing [10]. Coupled with these, it is common for both children and adults with DS to have difficulties with working memory, writing, and arithmetic [11,12], and for many people with DS, it is also associated with social withdrawal or poor social skills and in some cases insubordinate behaviour [13].

As a community, those with DS would appear to not meet the recommended minimums for physical activity of 150 min of moderate- or 75 min of vigorous-intensity exercise per week [14]. Recent studies suggest that they miss these targets by ~52% and ~74% for moderate and vigorous activity, respectively [15], with these low levels of activity being paired with a daily sitting time of 412.7 ± 216.6 min per day. Current studies from our group [9] demonstrated that through the application of just eight weeks of exercise in the form of walking, individuals with DS exhibited profound improvements in both physical and cognitive health. Indeed, 6 min walk distance (6MWT) increased by 11.4% coupled with improvements in vigilance measures ranging from 19.9 to 31.5%. Furthermore, selective attention as assessed using the STROOP test showed marked improvements across the eight weeks with the detection of correct responses increasing by 5.6% and incorrect responses decreasing by 37.9%. These findings suggest that walking acted as a modulator of cognitive processing through heightened arousal of the information processing cycle, eliciting increased vigilance and decision making.

The benefits of physical activity on mental health are well established in the general population. Indeed, exercise is routinely prescribed as an antidote to poor mental health and has become a central tenant of mental wellbeing recommendations across the world, with guidelines from the World Health Organisation (WHO) [16], American College of Sports Medicine (ACSM) [17] and Department of Health (DoH) in the UK [18] all citing evidence for the benefits of prescribed exercise. A recent prospective study of ~267,000 adults showed that higher levels of physical activity were associated with reduced odds of developing a mental health condition [19]. Exercise confers these benefits through a series of interconnected mechanisms relating to neurobiological adaptations [20,21] and cultivating behavioural change, related to self-regulatory skills and self-efficacy [22,23]. DS has previously been shown to be associated with low quality of life metrics and low self-efficacy, which is being associated with increased levels of anxiety, mood disturbance and symptoms related to depression [24]. However, there is limited evidence to show how prescribed exercise impacts self-esteem, quality of life, and mood states in populations with DS and much of which is restricted by limited sample sizes and controls. A comparative study examining the impact of competing in the Special Olympics on self-efficacy in DS showed a large effect size between competitors and matched non-competitors (ES = 1.36) [25]. Furthermore, Skotko and colleagues (2011) suggested that in a population with DS, the majority are actually happy with their lives. However, the authors offer a note of caution, stating that their sample was drawn from an affluent, mostly white population with higher functioning levels of DS [26]. Conversely, a Cochrane review highlighted the paucity of well-controlled studies examining the influence of prescribed exercise on both physical and psychosocial health in adults with DS, stating that of the 1954 articles identified, only 63 referred to exercise and DS, of which only 3 were deemed suitable for inclusion in the review [27]. Indeed, as Andriolo and colleagues (2005) highlighted, many studies are confounded by a lack of clinical measures, inadequate sample sizes and controls [27].

Therefore, the MinDSets study was undertaken to explore if the application of prescribed physical and/or cognitive exercise over an eight-week period could promote positive changes in satisfaction with life, self-efficacy, and mood states in adults with DS. It was hypothesised that physical activity would elevate perceptions of self-efficacy and adaptive mood states such as vigour in comparison to cognitive or combined training. It was further hypothesised that physical activity would temper perceptions of overall mood disturbance and counterproductive mood states such as anger, depression, and tension to a greater degree than those in the cognitive and combined groups.

## 2. Materials and Methods

This study was approved by the Anglia Ruskin University Research Ethics Committee (SES_Staff_19–25), with all data collection conducted in accordance with the guidelines established via the Declaration of Helsinki [28].

### 2.1. Study Design

Data presented in this study were collected as part of the MinDSets study [9]. Participants were placed in one of four groups based on their matched pre-intervention cardiorespiratory fitness test outcomes. All groups underwent an eight-week intervention period, which was preceded by a baseline assessment for physical and cognitive health and followed by a post-intervention assessment of those same measures. Participants in the exercise only group (EXE) were asked to perform 30 min of moderate-intensity cardiorespiratory exercise (either walking or jogging) three times a week for eight weeks. The cognitive training group (COG) completed eight levels (~20 min) of cognitive and executive function exercises six times a week through BrainHQ (Posit Science, San Francisco, CA, USA). The suite of games was selected after conducting a pilot study and was adjusted for each participant if the participant presented with a visual and/or auditory impairment [9]. This form of computerised training has previously been validated in a population with DS [29]. A third group of participants was instructed to implement the combined (COM) walking and cognitive training exercises for eight weeks, whilst a final group of participants was required to maintain their normal day-to-day-life and not partake in an intervention, therefore acting as a control group (CON) with a pseudo-intervention period of eight weeks.

### 2.2. Participants

Participants were recruited through a media campaign across Canada, North America, and Europe, and direct mailing from the Canadian Down Syndrome Society. Out of an initial 120 participants, 37 participants did not meet the inclusion criteria of being over the age of 18 years, being classed as ambulant, having no medical contraindication to exercise, and not having severe visual and/or auditory impairments that would prevent them from visualising information on computer and mobile/tablet screens or listening to instructions/auditory cues. Participants were also excluded if they did not have access to a mobile phone and computer, were not able to fulfil the time requirements of the study, and therefore had low adherence rates to the intervention programme, and if they stopped responding to the researchers. All participants had to have access to a helper or caregiver who would support them in their data collection. Eighty-three participants (40 females (F) and 43 males (M)) with an average age of 27.1 ± 8.0 years from ten countries across five continents (North America, *n* = 67; Europe, *n* = 8; Africa, *n* = 5; Asia, *n* = 2; and Australia, *n* = 1) provided informed consent and completed the study requirements. Twenty-two participants were assigned to EXE (F = 13, M = 9), 19 were allocated to COG (F = 15, M = 4), and COM (F = 7, M = 14) and CON (F = 5, M = 16) both had 21 participants, respectively. Participant recruitment was completed over a period of 12 months and data collection was conducted over 16 months to ensure that an estimated minimum group sample size of 20 participants was met, which was established through an a priori power calculation utilising previously established criteria for walking interventions in the DS population assuming a power (β) of 80% and an alpha level of 0.05.

### 2.3. Measurements

As part of the cognitive health assessments at baseline and after the eight-week intervention period through the online platform PsyToolkit [30,31], participants were asked to complete questionnaires to assess the satisfaction with life scale (SWLS) [32] and their generalised self-efficacy scale (GSE) [33,34]. Short-term mood states were assessed using an online version of the 65 item Profile of Mood States (POMS) instrument administered through Jisc Online Surveys. Participants were instructed to complete the POMS with their helper/caregiver present for when explanations regarding the different items were required in case of uncertainties. The POMS was completed at baseline, at the start of each week of the intervention as well as during the post-intervention assessment. Changes in estimated cardiorespiratory fitness from pre- to post-intervention were assessed through a modified 6 min walk test (6MWT), adapted from Chen et al. (2018) [35], which was recorded with a Fitbit Inspire 2 according to the methods of Merzbach et al. (2023) [9]. 

### 2.4. Data Acquisition

For the SWLS, a score for the five-item scale was calculated for each participant for baseline as well as post-intervention based on the possible responses from “strongly disagree” = 1, “disagree” = 2, “slightly disagree” = 3, “neither agree nor disagree” = 4, “slightly agree” = 5, “agree” = 6 to “strongly agree” = 7. A minimum score of five shows an extreme dissatisfaction with life, whilst the maximum score of 35 indicates that the participant is extremely satisfied with life. The change in SWLS (ΔSWLS) was calculated as the post-intervention score minus the baseline. The outcome scores of the ten-item GSE were calculated according to the responses of “not at all true” = 1, “hardly true” = 2, “moderately true” = 3 to “exactly true” = 4, with the change in GSE (ΔGSE) being the difference between post-intervention to pre-intervention. Scores for the mood states of tension, depression, anger, fatigue, confusion, vigour, friendliness, and total mood disturbance (TMD) from the 65 item POMS were determined from the five-point scale of “not at all” = 0, “a little” = 1, “moderately” = 2, “quite a bit” = 3 to “extremely” = 4 according to the methods of Kuesten et al. (2017) [36]. Three time points of the POMS were used for the final analysis: pre-intervention, mid-intervention, which is the average score from the week four and five time points, and post-intervention, with the change in each mood state (Δtension, Δdepression, Δanger, Δfatigue, Δconfusion, Δvigour, Δfriendliness, and ΔTMD) calculated from post- to pre-intervention. A bespoke app (MinDSets, Bliss Innovative Maker Studio, Brussels, Belgium), accessible via a browser on a mobile phone, was used to log the physical activity data recorded through personal Fitbit accounts and the cognitive training conducted through BrainHQ accounts for all participants. Adherence to the training interventions was monitored by a research assistant, and participants were encouraged on a weekly basis to complete the required walking and/or cognitive training sessions. The change in fitness (Δ6MWT) was determined as the difference between the distance covered in the 6MWT post-intervention minus the distance reached in the 6MWT pre-intervention.

### 2.5. Statistical Analysis

Statistical analysis was completed in SPSS (IBM SPSS Statistics, Version 29.0.0.0 (241), Armonk, NY, USA) and data are expressed, where appropriate, as the mean ± SD. Differences between groups for baseline or post-intervention were determined with independent samples Kruskal–Wallis tests utilising Bonferroni correction for multiple tests within the post hoc analysis. Responses from all participants independent of their group between time points were analysed using Wilcoxon signed rank tests. Time-by-group interactions across measures of mood states and groups were assessed through related samples Friedman’s Two-Way ANOVA by ranks tests with pairwise comparisons adjusted by the Bonferroni correction for multiple tests. Correlational analysis was conducted using Spearman’s rho test, with partial correlational analysis utilised for EXE and COM, taking the significant change in fitness [9] into account. An alpha level of *p* < 0.05 was accepted as statistically significant.

## 3. Results

### 3.1. Satisfaction with Life Scale and Generalised Self-Efficacy Scale Questionnaires

SWLS and GSE questionnaires were completed by all participants (*n* = 83) pre- and post-intervention. Table 1 shows the responses for both the SWLS and GSE questionnaire for the whole cohort (total) and split by groups for pre- and post-intervention. Overall, the participants were satisfied, indicated by a score of 26–30, or extremely satisfied (31–35) with life [32]. There were non-significant differences between groups at baseline (*p* > 0.05) and post-intervention (*p* > 0.05) and non-significant increases for SWLS scores from pre- to post-intervention in all responses and groups (*p* > 0.05).

Overall, participants scored their GSE below the international average of 29.55 [34] at pre-intervention but scores improved significantly to above the international average post-intervention with a change of 1.9 ± 5.2 (Z = −3.032, *p* = 0.002) for the whole group (Table 1). At baseline, there was a significant difference between groups (H(3) = 8.285, *p* = 0.04), with post hoc analysis showing that participants in EXE scored their self-efficacy significantly higher than those in CON (*p* = 0.030). Friedman’s Two-Way ANOVA by ranks showed non-significant differences from pre- to post-intervention in all groups (*p* > 0.05).

### 3.2. Profile of Mood States Questionnaire

POMS questionnaire completion rates ranged from 90 to 95% for the weekly recordings during the intervention, with all participants having completed the questionnaire at baseline. There was a 99% completion rate at the mid-point of the intervention as well as only one questionnaire not being completed post-intervention (*n* = 82).

#### 3.2.1. Overall Group Responses for POMS Outcomes over Time

Table 2 presents the overall responses for the POMS questionnaire mood states from pre- to mid- to post-intervention. Related samples Friedman’s Two-Way ANOVA by ranks showed that there were significant positive changes in mood states for the full cohort for tension (F(2) = 17.184, *p* < 0.001), with post hoc analysis revealing that participants felt significantly less tense from pre- to post-intervention (*p* < 0.001). Depression scores were significantly reduced (F(2) = 24.680, *p* < 0.001), with the decrease shown in the post hoc analysis from the pre- to post-intervention (*p* = 0.004) and mid- to post-intervention (*p* = 0.004) time points. Furthermore, participants reported a significant reduction in anger scores (F(2) = 13.917, *p* < 0.001) from pre- to post-intervention (*p* = 0.004). Fatigue scores were also significantly reduced (F(2) = 13.663, *p* = 0.001) from baseline to after the eight-week intervention period (*p* = 0.004). Confusion scores significantly improved (F(2) = 17.007, *p* < 0.001) from baseline to the mid-intervention point (*p* = 0.050) and further positive changes from the first questionnaire to the final questionnaire recording (*p* < 0.001). Non-significant changes were reported for vigour scores (*p* > 0.05). A slight negative reduction in friendliness outcomes (F(2) = 7.121, *p* = 0.028) was reported from pre- to mid-intervention only (*p* = 0.040). TMD was positively affected (F(2) = 13.473, *p* = 0.001) from pre- to post-intervention (*p* = 0.001) and mid- to post-intervention (*p* = 0.025).

#### 3.2.2. Between Group Differences for POMS Outcomes

Independent samples Kruskal–Wallis tests revealed significant differences between groups for depression scores at the mid-point of the intervention (H(3) = 8.542, *p* = 0.036), with post hoc analysis highlighting that participants in EXE were scoring significantly lower for depression than those in CON (*p* = 0.037) (Figure 1), and for confusion post-intervention (H(3) = 10.444, *p* = 0.015), with post hoc analysis showing lower confusion scores for EXE compared to CON (*p* = 0.022) (Figure 2). All other differences in mood states between groups were non-significant (*p* > 0.05).

#### 3.2.3. Within Group Differences for POMS Outcomes over Time

Friedman’s Two-Way ANOVA by ranks revealed the following differences in mood states within groups between time points. Tension showed a significant reduction in COG (F(2) = 8.771, *p* = 0.012) with post hoc pairwise comparison showing significantly lower tension scores from pre- to post-invention (*p* = 0.014) (Figure 3B). Decreases in tension over time were non-significant for the other three groups (Figure 3A,C,D).

There was also a significant change in depression scores in COG (F(2) = 10.548, *p* = 0.005); pairwise comparison highlighted a significant decrease from pre- to post-intervention (*p* = 0.011) (Figure 4B). CON also showed a significant change in depression scores (F(2) = 7.897, *p* = 0.019); however, post hoc analysis did not reveal a difference between time points (*p* > 0.05) (Figure 4D). No further significant changes in depression scores were found in the other groups (Figure 4A,C).

There were non-significant changes in anger scores within all groups across time points, which are presented in the Appendix A Figure A1. There was a significant difference in fatigue scores for COG (F(2) = 7.000, *p* = 0.030), but post hoc analysis could not confirm this change (*p* > 0.05). Fatigue scores for all groups over time are presented in the Appendix A Figure A2.

Confusion scores for EXE also indicated an overall reduction (F(2) = 6.256, *p* = 0.044), but this could not be confirmed through post hoc analysis (*p* > 0.05) (Figure 5A). There was a decrease in confusion scores for COG (F(2) = 13.178, *p =* 0.001), which was significant from pre- to mid-intervention (*p* = 0.017) and pre- to post-intervention (*p* = 0.003) (Figure 5B). Neither participants in COM (Figure 5C) nor CON (Figure 5D) reported significant changes in their confusion scores over time.

There were non-significant changes in vigour as well as friendliness within all groups across time points. Data are presented in the appendix for vigour (Figure A3) and friendliness (Figure A4).

TMD showed a significant change in COG (F(2) = 8.842, *p* = 0.012), which was due to a significant reduction in TMD scores from pre- to post-intervention (*p* = 0.011) (Figure 6B). There were non-significant changes in TMD scores for EXE (Figure 6A), COM (Figure 6C), and CON (Figure 6D).

### 3.3. Correlational Analysis

Spearman correlational analysis within groups between SWLS and GSE change scores (ΔSWLS and ΔGSE) to mood state changes revealed several significant correlations. For EXE, ΔSWLS was significantly negatively correlated to ΔTMD (r_S_ = −0.581, *p* = 0.006) (Figure 7A). This was explored further with a significant strong negative correlation highlighted between ΔSWLS and Δtension (r_S_ = −0.609, *p* = 0.003), and moderate negative correlations to Δanger (r_S_ = −0.494, *p* = 0.023), Δfatigue (r_S_ = −0.439, *p* = 0.046) and Δconfusion (r_S_ = −0.505, *p* = 0.020) (Figure 7B–E, respectively). Significant negative correlations between ΔSWLS and ΔTMD (r(18) = −0.531, *p* = 0.016), and Δtension (r(18) = −0.531, *p* = 0.016), and Δfatigue (r(18) = −0.475, *p* = 0.034) for EXE were confirmed through partial correlational analysis, which controlled for fitness change (Δ6MWT distance). There were non-significant interactions across all variables for COG (*p* > 0.05).

For COM, ΔGSE was significantly positively correlated with Δvigour (r_S_ = 0.779, *p* < 0.001) and Δfriendliness (r_S_ = 0.573, *p* = 0.007) (Figure 8A and Figure 8B, respectively). These correlations were again confirmed through partial correlational analysis controlling for the change in fitness (Δ6MWT distance) and revealed a strong positive correlation between ΔGSE and Δvigour (r(18) = 0.818, *p* < 0.001) and a moderate positive correlation to Δfriendliness (r(18) = 0.552, *p* = 0.012). There were non-significant correlations for all variables in CON (*p* > 0.05).

## 4. Discussion

The aim of this study was to address the limitations of previous studies [27] and examine the effect of chronic prescribed physical activity over an eight-week period on general self-efficacy (GSE), satisfaction with life (SWLS) and mood states (through POMS and TMD) in adults with DS. It was hypothesised that physical activity would elevate perceptions of GSE, SWLS and adaptive mood states such as vigour and attenuate perceptions of overall mood disturbance and specific maladaptive mood states such as anger, depression, and tension. Overall, the findings highlight the potential worth and utility of a prescribed chronic exercise intervention on promoting adaptive affective states and attenuating maladaptive affective states in a population with DS.

### 4.1. Life Satisfaction and Self-Efficacy

In this study, within-cohort and between-group differences in SWLS and GSE at two time points, pre- and post-intervention, were analysed. At baseline, and in line with previous work in this domain [26], all participants indicated they were satisfied (score of 26–30) or extremely satisfied (score of 31–35) with life, but there were no statistically significant changes in SWLS from pre- to post-intervention in the total sample or within any of the groups. GSE was below the international average of 29.55 [34] at baseline, but scores in the total sample improved significantly to above average throughout the duration of the study. However, pre- to post-intervention within-group changes in GSE were statistically non-significant. Between-group differences in POMS and TMD were analysed at three time points throughout the course of the study: pre-intervention, mid-intervention, and post-intervention. Between-group analyses indicated that individuals in the EXE group reported significantly lower perceptions of maladaptive mood states such as mid-intervention depression and post-intervention confusion in comparison to the CON group. In addition, within-group analyses were conducted to explore changes in POMS and TMD pre-, mid-, and post-intervention. Two-way ANOVAs revealed that only in the COG group tension, depression, confusion, and TMD were reduced between time points, although a similar pattern approached significance for attenuated fatigue for the COG group and reduced confusion in the EXE group.

Overall, the findings reflect positive albeit statistically non-significant changes in SWLS and GSE within the groups. Although contrary to the hypothesis, when grounded in theory and the previous literature, the results are perhaps unsurprising and further contribute to ambiguous findings on the effects of physical activity and exercise on SWLS and GSE specifically in people with DS. On a theoretical level, SWLS has been shown to have both trait and state-like properties [37], which are less sensitive to situational changes than mood states [38]. Consequently, to observe meaningful changes in trait-like characteristics associated with SWLS, longer-term interventions which adhere to self-determination principles of autonomy, relatedness, and competence to cause statistically meaningful change, may be more appropriate [39]. In the able-bodied population, robust evidence exists of the role acute and chronic exercise interventions play in enhancing SWLS and GSE [40,41]. For instance, at the acute intervention level, Yang et al. (2014) identified significant increases in self-efficacy and life satisfaction in older women following a 13 day lower extremity intervention [42]. Furthermore, at the chronic intervention level, Neupert et al. (2009) reported that after a six-month training intervention programme, healthy older adults reported higher levels of self-efficacy [43], and Vurgun (2015) reported that a 14 week aerobic training programme yielded significant improvements in satisfaction and self-efficacy in healthy middle-aged women [44].

However, findings specific to the population with DS are more ambiguous. A recent systematic review [45] summarised the effect of exercise interventions on daily life activities, including SWLS and GSE, and found that in a total of 33 outcome measures, only 19 (58%) yielded positive affective changes, with just 15 reaching statistical significance for SWLS and GSE. For instance, Heller et al. (2004) found that individuals with DS who participated in an exercise health education programme, three days per week for 12 weeks, reported larger GSE and SWLS outcomes than a control group [24]. Conversely, Shields and Taylor (2015) found no statistically significant changes in life satisfaction in young adults with DS following an eight-week intervention, which encouraged participants to walk for 150 min per week [46]. Consequently, the current findings may simply reflect ambiguous findings within the literature, which has examined the efficacy of exercise interventions, and may have been a product of the nature and characteristics of exercise interventions in able-bodied individuals and characteristics specific to the DS population [45,47]. It is noteworthy, however, that participants in the current study reported relatively high SWLS and GSE scores at baseline, and thus positive affective changes may not have been as incremental in comparison to physiological changes due to observed ceiling effects attributable to already positive mood states.

### 4.2. Between Group Findings for POMS Outcomes

The between-group findings that a prescribed chronic exercise intervention promotes adaptive affective states, such as positive changes to depression and confusion, are unsurprising, as observed in the current study in comparison to the CON group, when considered in the context of contemporary neurological and exercise psychology theory. On a neurological level, exercise and physical activity result in a corresponding increase in neurotransmitter activity [48], systematically increasing dopamine receptor availability and subsequently attenuating the severity of maladaptive mood states such as depression, anxiety, and confusion [49,50]. In addition, the Dual-Mode Model [51] for understanding affective responses to exercise of different intensities specifies that during moderate-intensity exercise, cognitive and interoceptive factors are unimportant determinants of affective responses to exercise as lower intensity exercise is no threat to homeostasis [52]. Consequently, moderate-intensity aerobic exercise, as prescribed to participants in this study, is experienced as pleasant and yields adaptive affective states [53]. 

The findings about the adaptive affective changes attributed to a prescribed chronic exercise programme in a population with DS are also consistent with research conducted on able-bodied individuals [40,54]. For instance, Steptoe et al. (1989) assigned adults from the general population to a 10-week moderate-intensity intervention programme and found significant reductions in tension, anxiety, and depression [55]. In addition, the current results are consistent with prescribed intervention-based studies conducted with an elderly population [56], individuals with a psychological disability (e.g., bipolar disorder) [57], and individuals with a physical disability (e.g., multiple sclerosis) [58]. For example, Ng et al. (2007) examined the effectiveness of a 40 min walking intervention at a self-determined intensity up to five times a week and found reductions in depression and anxiety in participants with psychiatric disorders [59]. Furthermore, Ensari et al. (2016) found that 20 min of treadmill walking led to reductions in TMD, anger, confusion, depression, and tension and improvements in vigour in 24 individuals with multiple sclerosis [58]. Collectively, the findings support neurological theory and the existing exercise psychology literature and tentatively suggest that the positive affective changes attributed to exercise and physical activity hold true during a chronic prescribed exercise programme in a population with DS. 

### 4.3. Within Group Findings for POMS Outcomes

A finding worthy of discussion is that within-group analyses revealed statistically significant reductions in maladaptive mood states, specifically tension, depression, confusion, and TMD between time points in the COG group. Furthermore, attenuations in fatigue were visible in the COG group, reductions in confusion were present in the EXE group and a decrease in depression was noted in the CON group; however, these changes were not statistically significant. All other within-group analyses comparing difference across time in POMS and TMD data were statistically non-significant. Taken together, it would appear that adaptive changes to maladaptive mood states occurred over time within the COG group alone and not the EXE and COM groups as anticipated. 

The positive effects on mood associated with the COG group appear to be consistent with previous studies. A recent systematic review [60] summarised the efficacy of commercially available “brain training” programmes such as BrainHQ, as used in this study. They reported small and positive effects on cognitive and executive functioning, fluid intelligence, memory, and attention in healthy and cognitively impaired older adults [61,62,63]. Interestingly, Nguyen et al. (2022) also found small and statistically significant mean effect sizes in favour of “brain training” compared to controls for everyday functioning including markers of mood and wellbeing [60]. For instance, Goghari and Lawlor-Savage (2018) instructed 97 healthy older adults to complete eight weeks of cognitive training and observed that mood significantly improved [64]. In addition, the feasibility of a visuospatial training intervention has also shown to be highly efficacious for children aged 7–12 years with DS on cognitive markers of healthy functioning like short-term memory [65]. Although in its infancy, recent studies tend to attribute “brain training” programmes with positive affective and cognitive consequences [66], as seen in the current study. 

Perhaps surprising are the findings that the COM and EXE groups did not experience statistically significant positive affective changes in response to their exercise intervention. These findings appear to be contradictory to evidence within able-bodied participants where reviews consistently highlight the efficacy of exercise and physical activity as determinants for adaptive affective change [67,68,69,70]. Furthermore, intervention programmes which combine “brain training” and physical activity also yielded promising findings concerning adaptive mood states. Specifically, individuals assigned to a physical and cognitive exercise group experienced significantly greater adaptive mood changes than participants in physical exercise or cognitive exercise groups alone [71]. Consequently, our findings that EXE and COM did not yield positive affective change appear to be contradictory to those in the able-bodied population. Indeed, in the COM group, it appears that the addition of exercise eliminated affective benefits to depression and confusion.

One potential explanation is that exercise and physical activity hold an attenuated effect on mood in the DS population compared to their able-bodied counterparts. In short, exercise is not as key a determinant to adaptive mood states in a population with DS. However, research assessing the effect of exercise upon mood with participants with other intellectual disabilities [72] reveal physical, cognitive, and affective health enhancements following a one-time running exercise at a self-selected pace. Rather, it is more likely that the nature of the exercise intervention may require more nuanced prescription whilst considering FITT principles, and exercise characteristics contextually relevant to a population with DS to optimise the efficacy of exercise interventions for enhancing mood in individuals with DS. For instance, walking applies little cognitive load in an able-bodied population, but the cognitive demands of walking are higher for the DS population [9], and it could be that the elevated cognitive demands and increased complexity associated with walking, as prescribed in this study, may have dampened anticipated affective mood consequences. One possible alternative exercise intervention strategy worthy of researchers’ efforts is Assisted Cycle Therapy (ACT). Previous studies have demonstrated that acute and chronic ACT yielded improvements in affective markers of well-being such as depression and cognitive markers of healthy functioning like cognitive planning and executive functioning [73]. A potential advantage of ACT in comparison to walking interventions when applied to a population with DS is predicated on the use of an augmented motor to aid participants to maintain a selected cadence, thus reducing the cognitive effort needed, consequently requiring the allocation of less resources to focus on achieving an assigned or selected cadence. It is noteworthy that the addition of exercise to the COG group, as required of the COM group, eliminated affective benefits to depression and confusion. It is possible to speculate that the addition of a second relatively cognitively demanding task, as required of the COM group, may have depleted already attenuated finite resources which reversed positive affective changes to confusion and depression.

Operational decisions associated with the prescribed exercise intervention may also explain the current findings. It is noteworthy that research, which has assessed the effect of exercise on mood in individuals with intellectual disabilities [72], required participants to self-select their exercise intensity, and it may be that an elevated sense of autonomy provided more intrinsic motivation which fostered more adaptive affective states [39]. Consequently, the prescriptive nature of the intervention may have acted as a buffer to the adaptive affective effects commonly attributed to exercise interventions. Furthermore, in the current study, participants in the COM and EXE groups completed exercise alongside their caregiver who may have engaged them in conversation. It is conceivable that an increase in cognitive load associated with walking and conversing may have burdened positive affective changes. For instance, healthy young adults are sufficiently skilled to feasibly allocate resources between speech and walking [74], whereas individuals with intellectual disabilities find allocating resources to both tasks more cognitively demanding [75].

### 4.4. Limitations and Future Research

There are limitations associated with this study which warrant acknowledgement and may inspire avenues for future work. POMS is a multi-dimensional construct that requires the participant to reflect on their mood, using a variety of statements against a five-point scale. The phrasing in a number of these statements could be considered nebulous and difficult to relate to, and responding to this might be conceivably more demanding for those with DS who may have intellectual difficulties. This resulted in caregivers having to assist with the completion of the survey and could, therefore, have created a bias in the outcomes of some of the responses. Similarly, the SWLS and GSE surveys were not designed to cater for those with learning disabilities and again required caregiver support in their completion.

The findings from this study pose questions for future work. Walking for people with DS has been defined as a skill [9], and it is contended that task complexity could render increases in cognitive development but, as is suggested, may dampen mood adaptation. Therefore, the challenge is to find the balance between task complexity, exercise intensity, and exercise load for people with DS. To the best of the authors’ knowledge, this is the first time the POMS questionnaire has been used in a population with DS. It is contended that future studies should consider modifying this template for use in all neurodivergent populations.

## 5. Conclusions

This is the first study of its kind to explore if the application of a period of prescribed physical activity and/or cognitive training can enhance measures of quality of life and traits of mood states in an adult community with DS. The findings are important and provide further evidence that physical activity can not only augment biological and cognitive health but may also enhance quality of life and mood. Additionally, the role of cognitive training or an imposed cognitive load should be recognised for again enhancing self-efficacy within adults with DS. The societal benefits are manifest for the community with DS; being more physically active and cognitively stimulated will promote enhanced self-worth and social integration.

## Figures and Tables

**Figure 1 ijerph-21-00610-f001:**
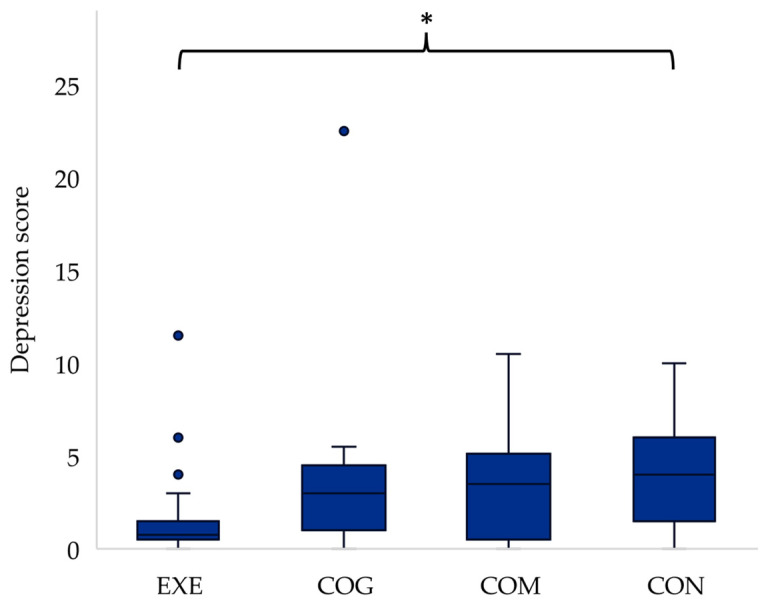
Between group differences in depression scores for the mid-intervention time point, with * indicating a significant difference between EXE and CON (*p* < 0.05). Outliers in the data are represented by dots outside of the whiskers on the graph.

**Figure 2 ijerph-21-00610-f002:**
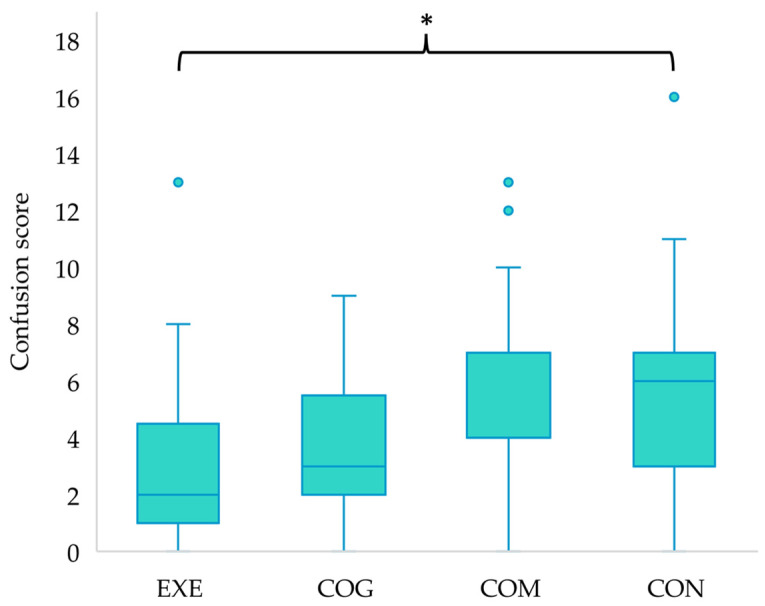
Between group differences in confusion scores for the post-intervention time point, with * highlighting a significant difference between EXE and CON (*p* < 0.05). Outliers in the data are represented by dots outside of the whiskers on the graph.

**Figure 3 ijerph-21-00610-f003:**
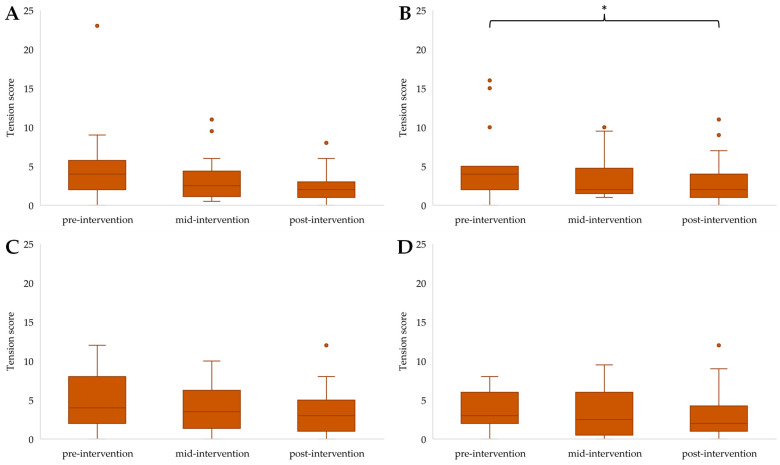
Tension scores for EXE (**A**), COG (**B**), COM (**C**), and CON (**D**) across time points, with * highlighting a significant reduction from pre- to post-intervention (*p* < 0.05). Outliers in the data are represented by dots outside of the whiskers on the graphs.

**Figure 4 ijerph-21-00610-f004:**
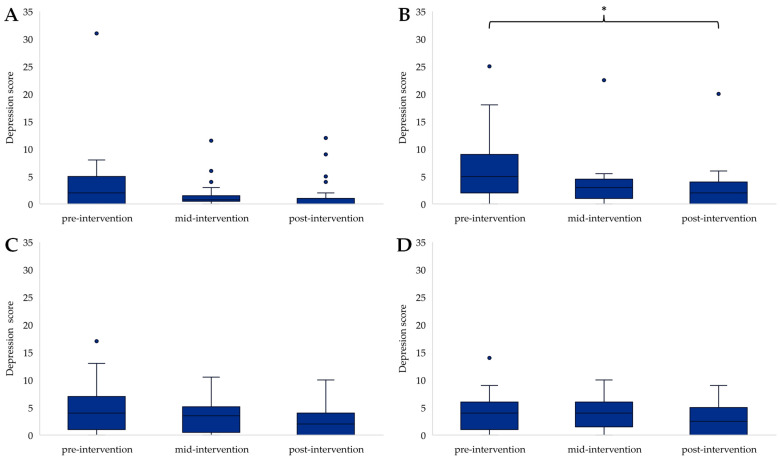
Depression scores for EXE (**A**), COG (**B**), COM (**C**) and CON (**D**) across time points, with * presenting a significant decline from pre- to post-intervention (*p* < 0.05). Outliers in the data are represented by dots outside of the whiskers on the graphs.

**Figure 5 ijerph-21-00610-f005:**
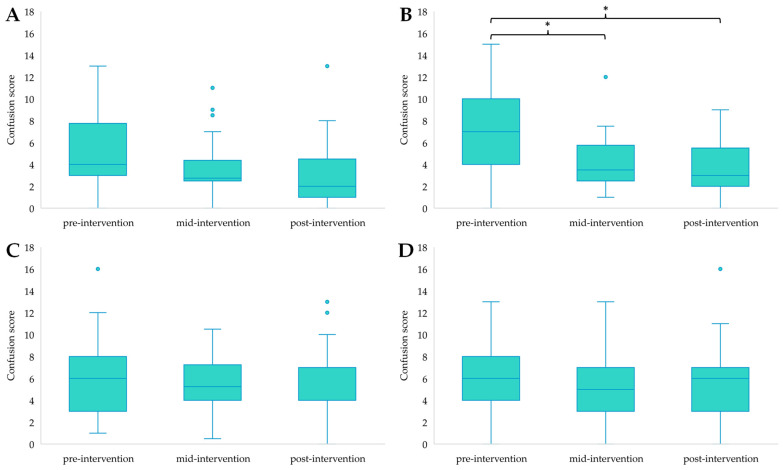
Confusion scores for EXE (**A**), COG (**B**), COM (**C**), and CON (**D**) across time points, with * highlighting a significant decrease from pre- to mid-intervention (*p* < 0.05) and pre- to post-intervention (*p* < 0.01). Outliers in the data are represented by dots outside of the whiskers on the graphs.

**Figure 6 ijerph-21-00610-f006:**
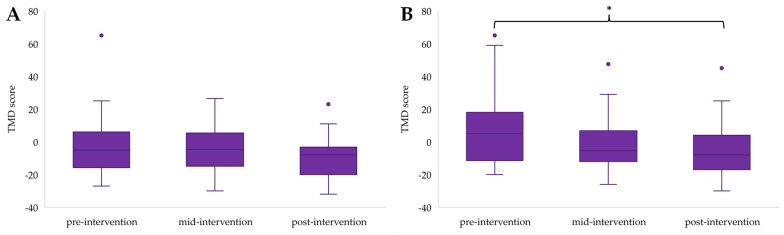
TMD scores for EXE (**A**), COG (**B**), COM (**C**), and CON (**D**) across time points, with * showing a significant reduction between pre- to post-intervention scores (*p* < 0.05). Outliers in the data are represented by dots outside of the whiskers on the graphs.

**Figure 7 ijerph-21-00610-f007:**
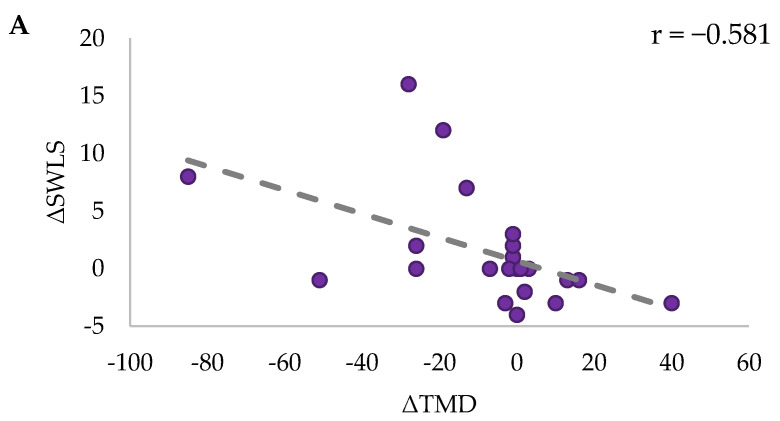
Correlations for EXE between ΔSWLS and ΔTMD (**A**), Δtension (**B**), Δanger (**C**), Δfatigue (**D**), and Δconfusion (**E**). R-values for each correlation are indicated in the top right corner of each graph.

**Figure 8 ijerph-21-00610-f008:**
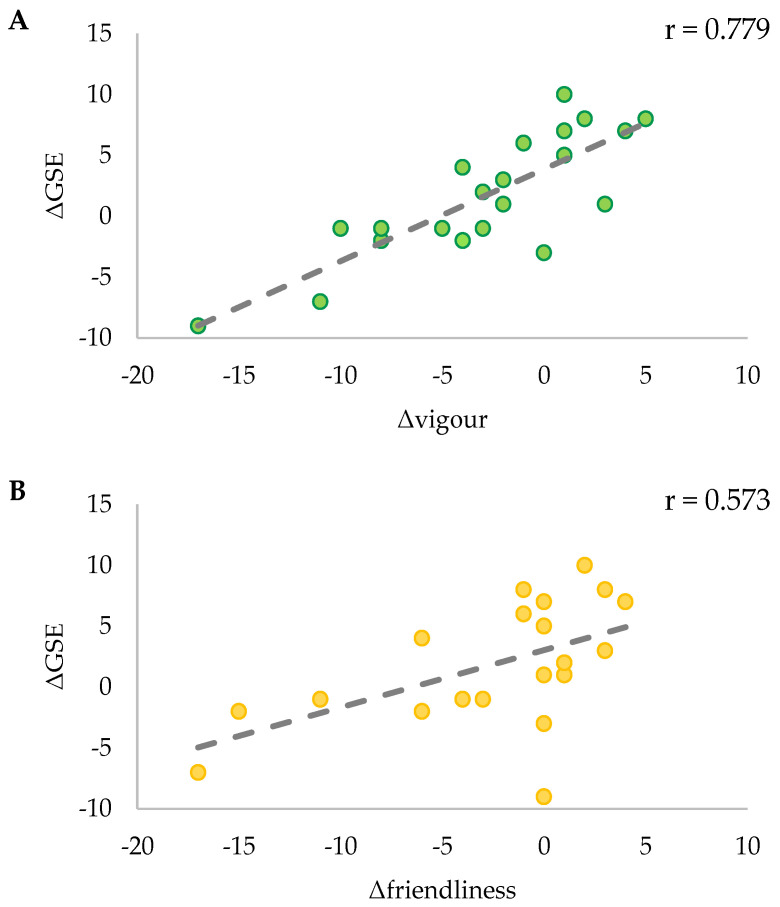
Correlations for COM between ΔGSE and Δvigour (**A**), and Δfriendliness (**B**). R-values for each correlation are indicated in the top right corner of each graph.

**Table 1 ijerph-21-00610-t001:** Outcomes of SWLS and GSE questionnaires from pre- to post-intervention for total responses and split by groups.

	SWLS			GSE		
	Pre	Post	Change	Pre	Post	Change
Total (*n* = 83)	30.1 ± 4.7	30.9 ± 3.9	0.8 ± 4.6	27.8 ± 5.9	29.7 ± 5.6	1.9 ± 5.2 ^a^
EXE (*n* = 22)	30.6 ± 4.8	32.1 ± 2.9	1.5 ± 5.0	30.3 ± 4.8 ^#^	31.2 ± 4.4	0.9 ± 4.9
COG (*n* = 19)	30.2 ± 5.4	30.4 ± 5.3	0.2 ± 5.1	27.6 ± 5.7	30.0 ± 5.8	2.4 ± 5.2
COM (*n* = 21)	30.4 ± 3.9	31.2 ± 3.3	0.9 ± 4.0	28.5 ± 5.2	30.1 ± 5.6	1.7 ± 5.1
CON (*n* = 21)	29.2 ± 4.6	29.9 ± 3.7	0.7 ± 4.6	24.8 ± 6.7	27.6 ± 6.7	2.8 ± 6.2

^a^ indicates a significant increase from pre- to post-intervention (*p* < 0.01); ^#^ highlights a significant difference to CON at pre-intervention (*p* = 0.03).

**Table 2 ijerph-21-00610-t002:** Changes in POMS questionnaire mood states from pre- to mid- to post-intervention for total responses.

	Pre-Intervention	Mid-Intervention	Post-Intervention
Tension	4.7 ± 4.0 ^a^	3.5 ± 3.0	2.9 ± 2.9
Depression	4.8 ± 5.6 ^a^	3.1 ± 3.5 ^c^	2.5 ± 3.5
Anger	3.4 ± 4.5 ^a^	2.0 ± 2.6	1.8 ± 2.6
Fatigue	3.9 ± 4.3 ^a^	3.1 ± 3.2	2.6 ± 3.7
Confusion	6.2 ± 3.8 ^a,b^	4.8 ± 2.9	4.5 ± 3.5
Vigour	19.9 ± 6.5	18.2 ± 6.9	18.9 ± 7.2
Friendliness	20.3 ± 4.9 ^b^	19.0 ± 5.6	19.1 ± 6.1
TMD	3.1 ± 21.0 ^a^	−1.7 ± 15.4 ^c^	−4.6 ± 15.9

TMD, total mood disturbance; ^a^ highlights a significant difference from pre- to post-intervention (*p* < 0.01); ^b^ presents a significant difference from pre- to mid-intervention (*p* < 0.05); ^c^ shows a significant difference from mid- to post-intervention (*p* < 0.05).

## Data Availability

Data available on request due to ethical restrictions. The data presented in this study are available on request from the corresponding author. The data are not publicly available due to ethical restrictions.

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
