# Peer review of "The Effects of Prescribed Physical and Cognitive Exercise on Life Satisfaction, Self-Efficacy and Mood States in Adults with Down Syndrome: The MinDSets Study"

_ijerph, 2024, doi:10.3390/ijerph21050610_

Round 1

Reviewer 1 Report

Comments and Suggestions for Authors

SUMMARY:  In this study, the authors analyze the psychosocial effects of an exercise and/or cognitive training program on 83 adults with Down syndrome.  The study is comprehensive, and I commend the authors for doing this much-needed research.  This manuscript is well written and contributes substantially to a parse literature on the topic.  Before this manuscript is accepted for publication, I would recommend that the authors consider the following points to only further strengthen their study.

ABSTRACT

Line 11:  The authors write, “Down Syndrome (DS) is characterised by a duplication of chromosome-21 and is linked to contraindications in both physical and mental health, including low self-efficacy and disturbed mood states.”  Some of the wording here is awkward and not accurate.  I would recommend instead, “…linked to co-occuring physical and mental health conditions.”

INTRODUCTION

Line 34:  The authors should provide a reference for incidence rate.  Additionally, they should cite the county since published incident rates vary for Down syndrome by country because of effectives of elective terminations.

Line 62:  The authors should provide the references for guidelines from WHO, ACSM, and NHS.

Line 66:  The authors write, “DS has previously been shown to be associated with low quality of life metrics and low self-efficacy, which is being associated with increased levels of anxiety, mood disturbance and symptoms related to depression.”   In contrast, the authors should cite and review this paper, which documents, in this cohort, that people with Down syndrome have positive moods and self-perception:

Skotko, B.G., Levine, S.P., Goldstein, R. (2011).  Self-perceptions from People with Down Syndrome.  American Journal of Medical Genetics, Part A: 155:2360-2369

Line 79: The authors should also briefly review the evidence of exercise on mitigating Alzheimer’s disease in the neurotypical population.  They should then briefly review the literature, demonstrating that adults with Down syndrome are more prone to developing Alzheimer’s disease at an earlier age.  To this extent, measuring an exercise intervention for this population might have potential benefits in reducing downstream neurological conditions such as Alzheimer’s disease.

METHODS

Line 97:  How did the authors measure fidelity to the exercise regimen in the EXE and Com groups?

Line 99:  The authors should describe in more detail what the COG training included.  What exactly did the participants do?

Line 127:  The authors should state and cite whether the instruments used were (a) previously validated and used in the neurotypical population and (b) previously validated and used in the Down syndrome population.

Line 130-131:   With this being a virtual study, how did the authors ensure that the person completing the SWLS and GSE was the participant with Down syndrome, and not the caregiver?  How did they ensure that the participant was not influenced or guided in responding by the caregiver?

RESULTS

Table 1:  I would recommend putting the superscript a next to the numbers of change, rather than pre, for the total group on GSE.  And, are the authors stating that only the GSE pre for EXE differed significantly from CON, or are they saying that the change for pre EXE differed significantly form the change for CON?

FIGURES 1 and 2:  These figures demonstrate that when the cognitive element was combined with the exercise component, the benefits to depression and confusion were eliminated.  Why do the authors think this is so?  They should elaborate on potential thoughts in the Discussion section.

Figures 1 and 2:  I would recommend that the authors combine these figures into a multi-panel figure and include the plots for the other measures of the POMS questionnaire (e.g., tension, etc.)  It will be equally important for the reader to see graphically those measure that had no statistical difference.

Figures 3, 4, 5, and 6:  I would similarly recommend that the authors combine these figures, all from COG group, into a multi-panel figure and include the plots for the other measures of the POMS questionnaire (e.g., tension, etc.)  It will be equally important for the reader to see graphically those measure that had no statistical difference.  In their supplementary materials, I would recommend that they include the same panel for the other groups: EXE, COM, and CON.  The main manuscript can just reference that there were no statistical differences in these groups.

DISCUSSION

Line 306:  Authors should add “at baseline” in “All participants indicated they were satisfied (score of 26–30) or extremely satisfied (score of 31–35) with life at baseline…”  The authors should reference that this is in alignment with previous work:

Skotko, B.G., Levine, S.P., Goldstein, R. (2011).  Self-perceptions from People with Down Syndrome.  American Journal of Medical Genetics, Part A: 155:2360-2369

Line 308:  I recommend that the authors re-word to read: “GSE was below the international average (29.55) [27], but scores improved significantly to above average across all study groups over time. That is, pre- to post-intervention within group changes in GSE were statistically non-significant.”  The authors just can’t say post-intervention, since the CON group improved over time and did not receive intervention.  If anything, the CON group improved the most, suggesting a testing effect over time or the fact that people with Down syndrome just get happier over time, in general.

Line 338:  The authors might consider adding that, perhaps, the difficulty in showing change in the SWLS and GSE in the Down syndrome population is because they already had high levels of happiness and self-satisfaction at baseline.  So, perhaps, they had already reached ceiling effect, making the contributions of exercise, while important physiologically, not as incremental to their already positive moods.

Line 391:  The authors might find it helpful to review this additional paper looking at the effects of brain training on patients with Down syndrome: https://pubmed.ncbi.nlm.nih.gov/23734613/

Line 460:  I would recommend that the authors change “can also enhance” to “may also enhance” since, they acknowledge, that some of the results are ambiguous and contradictory.

Reviewer 2 Report

Comments and Suggestions for Authors

Refer to attached report.
